Hepatic metabolite responses to 4-day complete fasting and subsequent refeeding in rats

Sui Xiukun 1 2 3
Wang Hailong 2
Wu Feng 2
Yang Chao 2
Zhang Hongyu 2
Xu Zihan 2
Guo Yaxiu 2
Guo ZhiFeng 2
Xin Bingmu 3
Ma Ting tma@hit.edu.cn 1
Li Yinghui 2
Dai Zhongquan daizhq77@163.com 2
1 Department of Electronic and Information Engineering, Harbin Institute of Technology at Shenzhen , Shenzhen , China
2 State Key Laboratory of Space Medicine Fundamentals and Application, China Astronaut Research and Training Center , Beijing , China
3 Space Science and Technology Institute , Shenzhen , China
Barnett Matthew
Electronic publication date: 2022 Sep 20
Publication date: 2022
Volume: 10
Electronic Location ID: e14009
Received 2022 Feb 8; Accepted 2022 Aug 15
Copyright: ©2022 Sui et al.
Copyright year: 2022
Copyright holder: Sui et al.
License: This is an open access article distributed under the terms of the Creative Commons Attribution License, which permits unrestricted use, distribution, reproduction and adaptation in any medium and for any purpose provided that it is properly attributed. For attribution, the original author(s), title, publication source (PeerJ) and either DOI or URL of the article must be cited.
License URL: https://creativecommons.org/licenses/by/4.0/

Keywords: Prolonged fasting, Gluconeogenesis, Lipolysis, Ketogenesis, Metabolic syndrome, Nutrient-sensing signaling molecules, Hormone

Funding: Shenzhen Science and Technology Innovation Commission 2020 Basic Research Project JCYJ20200109110630285 State Key Laboratory of Space Medicine Fundamentals and Application and the China Astronaut Research and Training Center SMFA17A02 SMFA18B02 SMFA18B06 SMFA19A01 SMFA19C03 Space Medical Experiment Project of China Manned Space Program HYZHXM01002 This research was funded by the Shenzhen Science and Technology Innovation Commission 2020 Basic Research Project (JCYJ20200109110630285); the State Key Laboratory of Space Medicine Fundamentals and Application and the China Astronaut Research and Training Center (SMFA17A02, SMFA18B02, SMFA18B06, SMFA19A01, SMFA19C03), and the Space Medical Experiment Project of China Manned Space Program (HYZHXM01002). There was no additional external funding received for this study. The funders had no role in study design, data collection and analysis, decision to publish, or preparation of the manuscript.

==============================
Background

Fasting has been widely used to improve various metabolic diseases in humans. Adaptive fasting is necessary for metabolic adaptation during prolonged fasting, which could overcome the great advantages of short-term fasting. The liver is the main organ responsible for energy metabolism and metabolic homeostasis. To date, we lack literature that describes the physiologically relevant adaptations of the liver during prolonged fasting and refeeding. For that reason, this study aims to evaluate the response of the liver of Sprague-Dawley (SD) rats to prolonged fasting and refeeding.

Methods

Sixty-six male SD rats were divided into the fasting groups, which were fasted for 0, 4, 8, 12, 24, 48, 72, or 96 h, and the refeeding groups, which were refed for 1, 3, or 6 days after 96 h of fasting. Serum glucose, TG, FFA, β-hydroxybutyrate, insulin, glucagon, leptin, adiponectin and FGF21 levels were assessed. The glucose content, PEPCK activity, TG concentration and FFA content were measured in liver tissue, and the expression of genes involved in gluconeogenesis (PEPCK and G6Pase), ketogenesis (PPARα, CPT-1a and HMGCS2) and the protein expression of nutrient-sensing signaling molecules (AMPK, mTOR and SIRT1) were determined by RT-qPCR and western blotting, respectively.

Results

Fasting significantly decreased the body weight, which was totally recovered to baseline after 3 days of refeeding. A 4-day fast triggered an energy metabolic substrate shift from glucose to ketones and caused serum hormone changes and changes in the protein expression levels of nutrient-sensing signaling molecules. Glycogenolysis served as the primary fuel source during the first 24 h of fasting, while gluconeogenesis supplied the most glucose thereafter. Serum FFA concentrations increased significantly with 48 h of fasting. Serum FFAs partly caused high serum β-hydroxybutyrate levels, which became an important energy source with the prolongation of the fasting duration. One day of refeeding quickly reversed the energy substrate switch. Nutrient-sensing signaling molecules (AMPK and SIRT1 but not mTOR signaling) were highly expressed at the beginning of fasting (in the first 4 h). Serum insulin and leptin decreased with fasting initiation, and serum glucagon increased, but adiponectin and FGF21 showed no significant changes. Herein, we depicted in detail the timing of the metabolic response and adaptation of the liver to a 4-day water-only fast and subsequent refeeding in rats, which provides helpful support for the design of safe prolonged and intermittent fasting regimens.

Introduction

Many people undergo periodic fasting for health, religious or cultural reasons. Fasting regimens have been demonstrated to potentially benefit various aspects of health. They have become a therapeutic strategy for modern chronic metabolic diseases, such as obesity, insulin resistance, type 2 diabetes mellitus (T2DM), and nonalcoholic fatty liver disease (NAFLD) (Costello & Schones, 2018). Indeed, different periodic fasting strategies, such as intermittent and alternate-day fasting, have been designed to reverse the features of metabolic syndrome in overweight and obese individuals (Halpern & Mendes, 2021; Razavi et al., 2021; Varady et al., 2009). However, their advantageous effects vary greatly because most of the easily acceptable fasting models were conducted within 12–48 h or less of complete fasting, which do not cause drastic alterations in the metabolic pattern but do cause changes in glycogen interchange. Prolonged fasting represents an even more significant energy status perturbation (Horton & Hill, 2001). Previous studies have shown that three sequential metabolic phases occur during prolonged fasting in humans, nonhibernating mammals, and birds (Bertile & Raclot, 2008; Groscolas & Robin, 2001; Owen et al., 1998). Ample evidence also indicates that metabolic responses induced by short-term fasting might differ substantially from those caused by prolonged fasting (Klain et al., 1977). Interpretation of these influences requires an in-depth understanding of the changes in metabolite flux and the choice of a suitable timing pattern for fasting and refeeding.

Fasting is an adaptive metabolic process that occurs when exogenous nutrient intake is limited (Longo & Mattson, 2014). Adapting to prolonged fasting involves a coordinated series of metabolic shifts (Ibrahim et al., 2020). There are two different regulatory processes, one homeostatic and one circadian, induced by fasting (Escobar et al., 1998). The first process is aimed at maintaining energy balance. Usually, fasting quickly mobilizes glycogenolysis and gluconeogenesis, accompanied by the initiation of adipose tissue lipolysis. Then, fatty acids are released and converted into ketone bodies to provide the main metabolic fuel for extrahepatic tissues after glycogen depletion (Petersen, Vatner & Shulman, 2017). With an increase in the fasting duration, primarily lipid fuels are exhausted, and whole-body proteins are utilized for energy supply, which is usually considered to reflect a degenerating and irreversible state (Kamata et al., 2018; Kramer et al., 2018). The second process is manifested by the persistence of temporal fluctuations in triacylglycerides, free fatty acids, glucose, and proteins (Escobar et al., 1998). Refeeding stimulates hepatic glucose uptake, restores glycogen stores, and induces fatty acid synthesis, thereby inhibiting β-oxidation and ketogenesis (Cassidy et al., 2018; González-Casimiro et al., 2021). Several studies have proven that refeeding could lead to the normalization of resting energy expenditure and protein synthesis recuperation in humans and animals. Refeeding, especially after prolonged fasting, could also cause refeeding syndrome, which has been proven to be lethal (Ponzo et al., 2021; Runde & Sentongo, 2019). Therefore, an adequate and appropriate refeeding cycle is essential.

As a key hub that maintains metabolic homeostasis during fasting and refeeding (Rui, 2014), liver energy metabolism is tightly regulated by circulating hormones and nutrient signals, such as insulin (Boland et al., 2018), glucagon (Ramnanan et al., 2011), fibroblast growth factor 21 (FGF21) (Szczepańska & Gietka-Czernel, 2022), leptin (Zieba, Biernat & Barć, 2020), and adiponectin (Koch et al., 2014). Serum insulin and leptin decrease during short-term fasting. They play a pivotal role in the neuroendocrine response to fasting. Adiponectin levels are elevated during fasting, which stimulates food intake and decreases energy expenditure (Yanai & Yoshida, 2019). Glucagon promotes hepatic glucose production, thereby maintaining glucose homeostasis in the fasting state (Yang et al., 2019). FGF21 is a metabolic regulator of energy balance, lipid metabolism and glucose metabolism, and increased by fasting (Szczepańska & Gietka-Czernel, 2022). Many experiments have shown that circulating hormonal levels constantly change and play an important role in adjusting the metabolic response in short-term fasting. However, the responses of different energy-related hormones to the stress of prolonged fasting/refeeding have not yet been characterized. Additionally, imposing refeeding on this metabolic environment may result in very different nutrient and hormone responses relative to short-term fasting. Therefore, investigation of the different hormonal responses to prolonged fasting and subsequent refeeding is essential for the safety of prolonged intermittent fasting.

Accordingly, we conducted a 4-day complete prolonged fasting and subsequent refeeding study in rats to clarify the timing of the hepatic metabolic adaptations. We focused on serum hormones and metabolites, hepatic glycogen and triglyceride contents, the activities of rate-limiting enzymes in gluconeogenesis and ketogenesis, the protein expression of nutrient-sensing signaling molecules, and the mRNA and protein expression levels of enzymes that regulate flux through β-oxidation, ketogenesis, and gluconeogenesis during different fasting and refeeding durations. Together, these studies define the timing of the changes in hepatic metabolism during prolonged fasting and provide data to design an effective intermittent prolonged fasting regimen for metabolic disease improvement.

Material and Methods

These studies were approved by the Committees of Animal Ethics and Experimental Safety of the China Astronaut Research and Training Center, and all procedures were performed in accordance with the Animal Ethics Council (ACC-IACUC-2021-030). Animals received humane care in accordance with the principles in the Guide for the Care and Use of Laboratory Animals protocol published by the Ministry of the People’s Republic of China (issued on June 4, 2004).

Animals and experimental design

Sixty-six healthy male Sprague–Dawley (SD) rats (Rattus norvegicus), twelve to fourteen weeks old and 250–270 g, were purchased from Beijing Vital River Laboratory Animal Technology Co., Ltd. (Animal quality certificate number: 110011210100824578) (Beijing, China). The rats were not genetically engineered, and no previous procedures were performed before the experiments. Each rat was considered an experimental unit. The rats were housed six per clean cage and exposed to a 12-h light/12-h dark cycle to adapt to the environment for one week before study initiation under appropriate humidity (45 to 55%) and ventilation conditions at room temperature (26  ± 1 °C). They were given ad libitum access to standard rodent chow and filtered water. No adverse events were observed.

For the prolonged fasting experiments, the rats were randomly divided into an ad libitum-fed group (n = 6) and groups that were fasted for different lengths of times (4 h, 8 h, 12 h, 24 h, 48 h, 72 h, or 96 h, n = 6 in each group). The rats were placed in an individual cage and exposed to a 12-h light/12-h dark cycle (lights on from 08:00 to 20:00 h) under appropriate humidity (45 to 55%) and ventilation conditions at room temperature (26  ± 1 °C) and were given only ad libitum water before sacrifice. Additionally, in the prolonged fasting period, when the rats had low blood glucose (<20 mg/dl) and high β-hydroxybutyrate levels (>4 mmol/L) or became sluggish with less physical activity, they were considered as an agonal status. At that point, the rats were anesthetized. In the refeeding study, rats were given access to 100 g of standard rat chow at 8:00–10:00 in the morning after 96 h of fasting. The control group was given ad libitum access to standard rodent chow at 8 o’clock in the morning every day. The food intake was assessed daily by weighing the unconsumed feed Before food replacement. All the rats were weighed at each fasting and refeeding duration time point.

To minimize potential confounders, all fasting rats were deprived of food beginning at the same time (8:00 am) in the same room during the experiments under the same lighting and temperature conditions. Equal amounts of food were given to the refeeding group. During the analysis of the biological samples, all of the investigators involved were blinded to the identity of the samples, except for those who then performed the statistical analysis.

Sample collection and storage

Rats were anesthetized from 8:00 am following different fasting duration (4 h, 8 h, 12 h, 24 h, 48 h, 72 h, and 96 h) and refeeding times (1 day, 3 days, or 6 days) with chloral hydrate and were sacrificed by cervical dislocation. Blood samples were collected from the heart. The collected blood was centrifuged at 3,000 × g for 10 min at 4 °C, and the supernatants were stored at −80 °C until analysis. The whole liver was quickly separated, the blood was cleaned from it in ice-cold physiological saline, and it was then blotted with filter paper, immediately frozen with liquid nitrogen, and stored at −80 °C until detection. We powdered a portion of the frozen liver using a liquid nitrogen-cooled mortar and pestle to obtain a homogenous liver sample.

Blood biochemical analysis

Blood glucose and β-hydroxybutyrate were determined by allowing a drop of tail vein blood to fall onto the test strip of a blood glucose or ketone monitor (FreeStyle Optium, Abbott, USA). Plasma free fatty acids (FFAs), insulin, glucagon, leptin, adiponectin, and FGF21 were determined by commercial enzyme-linked immunosorbent assay (ELISA) kits from Beijing Dongge Biotechnology Co., Ltd. (Beijing, China) following the manufacturer’s protocols. Triglyceride (TG) concentrations were measured with the Infinity Triglyceride Kit (Thermo Electron Corporation, Waltham, MA, USA) using a standard glycerol solution (Sigma, St. Louis, MO, USA).

ELISA

The glycogen content and PEPCK activity in powdered liver tissue were determined by commercial ELISA kits from Beijing Dongge Biotechnology Co., Ltd. (Beijing, China) according to the manufacturer’s protocols and are expressed per gram of tissue.

Quantitative real-time PCR (qPCR) assay

Total RNA was extracted from liver tissues by TRIzol (Invitrogen, Waltham, MA, USA) and reverse transcribed into cDNA by a commercial PrimeScript RT reagent kit (TaKaRa, Dalian, China) according to the manufacturer’s instructions. cDNA was used to detect mRNA expression by quantitative PCR using TB Green Premix Ex Taq II (Tli RNaseH Plus) (TaKaRa, Dalian, China). The primer sequences of 3-hydroxy-3-methylglutaryl coenzyme A synthase II (HMGCS2), peroxisome proliferator activated receptor α (PPARα), glucose 6-phosphatase (G6Pase), phosphoenolpyruvate carboxykinase (PEPCK), and carnitine palmitoyltransferase (CPT-1a), which are listed in Table 1, were synthesized by Genewiz (Suzhou, China). Real-time PCR was run on a Light Cycler 96 Real-Time system (Roche, Switzerland). The quantitative RT-PCR conditions were 95 °C for 5 min and 40 cycles of 95 °C for 30 s, 60 °C for 30 s and 72 °C for 30 s. The target gene expression was normalized to the housekeeping gene β-actin, and the relative mRNA levels of these genes were analyzed using the 2−ΔΔCt method (Livak & Schmittgen, 2001).

Western blot analysis

The liver tissues were homogenized in RIPA buffer with protease and phosphatase inhibitor cocktail (Thermo Fisher Scientific, Waltham, MA, USA) and then centrifuged at 12,000 rpm and 4 °C for 10 min. After adding loading buffer and boiling, the supernatants were collected and loaded onto sodium dodecyl sulfate–polyacrylamide gel electrophoresis (SDS–PAGE) gels. After running the gels, the proteins were transferred to a polyvinylidene difluoride (PVDF) membrane. Subsequently, the membranes were blocked with 5% skimmed milk for 1 h at room temperature. The following antibodies and concentrations were used overnight at 4 °C: PEPCK (1:1,000, Cell Signaling Technology, Danvers, MA, USA), HMGCS2 (1:1,000, Cell Signaling Technology, Danvers, MA, USA), AMPK (1:1,000, Cell Signaling Technology, Danvers, MA, USA), p-AMPK (1:1,000, Cell Signaling Technology, Danvers, MA, USA), mTOR (1:1,000, Cell Signaling Technology, Danvers, MA, USA), p-mTOR (1:1,000, Cell Signaling Technology, Danvers, MA, USA), and SIRT1 (1:1,000, Cell Signaling Technology, Danvers, MA, USA). The peroxidase-conjugated secondary antibodies were detected using enhanced chemiluminescence (Millipore Corp., Danvers, MA, USA). Densitometric analysis of the immunoreactive bands was performed using a chemiluminescence system (Tanon 5200, China). All target proteins were quantified by normalizing them to β-actin, which was reprobed on the same membrane and then used to calculate the quantity as a percentage of the control group.

Table 1 The primers list.

Gene name	Forward	Reverse	
PEPCK	GCAGAGCATAAGGGCAAGG	CAAAGAAGGGCCGCATAG	
G6Pase	CGGGAGGAGGGGGAGTGTTT	GCAGCGTGGTCAGGGAAGCA	
HMGCS2	CAGCTTACCGCAGGAAAATCC	CAAAAGGGTGTGTGGAAGATCA	
CPT-1a	CCCTAAGCCCACAAGGCTAC	AGCCTTTGCCGAAAGAGTCA	
PPAR α	TTGTGCATGGCTGAGAAGAC	CTGCCTCCTTGTTTTCAACG	
β-actin	ACGAGGCCCAGAGCAAGA	TTGGTTACAATGCCGTGTTCA	

Statistical analysis

The results are expressed as the mean ± standard deviation. All data were analyzed by one-way ANOVA or two-way ANOVA followed by Tukey’s multiple-comparison post hoc test. For the gene expression assessment, the 2−ΔΔCT values were analyzed. All statistical analyses were performed with Prism software (GraphPad Prism for Windows, version 8.0). P values < 0.05 were considered statistically significant.

Results

Body weight (BW) changes

All the rats had free physical activity and were alive throughout the fasting period. BW was recorded on the day of detection and is shown in Fig. 1. BW presented significant alterations at the different fasting time points, with losses of 6% at 4 h (n = 6), 5.3% at 8 h (n = 6), 7.8% at 12 h (n = 6), 15.3% at 24 h (n = 6), 18.7% at 48 h (n = 6), 25.5% at 72 h (n = 6), and 28.8% at 96 h (n = 6) compared with BW before food deprivation. Refeeding quickly restored the BW with a significant increase after 1 day of refeeding (n = 6), followed by continued growth in a time-dependent manner (p < 0.01). Over the 6 days of refeeding (n = 6), BW increased by 64.7% when compared with the 4-day fasting BW and was almost similar to the control group. The BW of the control group increased by 26.2% at the end of the entire experiment.

Figure 1 Body weight changes in response to fasting and refeeding.

Body weight was recorded before fasting (F0h) and fasting for 4, 8, 12, 24, 48, 72 or 96 h (Fxh), and after 1-, 3-, or 6-day refeeding (Rxd) following 96 h fasting. Data were presented as means ± SD (n = 6). Two-way ANOVA was used with multiple comparisons using post hoc Tukey’s test. F, Fasting; R, Refeeding. Significance was determined as *, #, $, p < 0.05; **, ##, $, p < 0.01, *, VS, respectively.

Serum metabolites in response to fasting and refeeding duration

The blood parameters of glucose, TG, FFAs, and β-hydroxybutyrate under fasting (n = 6 in each group) and refeeding conditions (n = 6 in each group) are summarized in Fig. 2. Blood glucose concentrations continued to decrease from 127.41 ± 8.78 mg/dl before fasting to the lowest level of 79.67 ± 4.46 mg/dl after 24 h of fasting and then slightly increased at 48 h and 96 h (Fig. 2A). At the end of the fasting experiment, the blood glucose exhibited an overall decrease of 29.9% compared with the baseline. Refeeding quickly reversed the effect of fasting on glucose, and serum glucose levels recovered to 139.2 ± 10 mg/dl after 1 day of refeeding and were close to the ad libitum levels at 3 days and 6 days of refeeding (128.3 ± 3.33 mg/dl and 126 ± 5.5 mg/dl, respectively). Serum TG (Fig. 2B) increased from 136.17 ± 32.92 mg/dL before fasting to 245.17 ± 45.73 mg/dL after 4 h of fasting and then markedly declined until the end of fasting. One day of refeeding increased serum TG from 16.0 ± 3.78 mg/dL after 96 h of fasting to 109 ± 28.08 mg/dL, but it was slightly decreased with additional refeeding time. In accordance with the increased lipolysis of adipose tissue, circulating FFAs (Fig. 2C) dramatically increased from 0.61 ± 0.14 µmol/L to 0.72 ± 0.11 µmol/L at 4 h of food deprivation, reached their peak value at 48 h (1.52 ± 0.22 µmol/L), and then slightly declined at 72 h and 96 h (1.1 ± 0.1 µmol/L and 0.86 ± 0.07 µmol/L, respectively). Refeeding quickly recovered the serum FFA levels to baseline, and they were significantly lower after 3 and 6 days of refeeding. The serum FFA mostly originated from the adipose tissue, so we also measured the contents of FFA in visceral and subcutaneous adipose tissue. In visceral adipose, FFA concentration increased from 1.67 ± 0.15 µmol/g to 1.83 ± 0.13 µmol/g after 4 h of fasting (Fig. S2A). It continued to increase in the subsequent fasting period, reached its highest at 24 h (2.17 ± 0.13 µmol/g), then visceral FFA began to decline from 48 h to 96 h of fasting. Refeeding reversed the trends of decline, and it returned to the base line level after 3 days of refeeding. Similar to visceral, the content of FFA in subcutaneous adipose tissue quickly increased from 1.57 ± 0.15 µmol/g to 1.73 ± 0.19 µmol/g after 4 h of fasting, and significantly increased at 8 h of fasting (1.97 ± 0.18 µmol/g, p < 0.01). It remained at high level until 24 h of fasting, then was declined at 48 h and 96 h (Fig. S2B). The most robust response to fasting was the increase in serum β-hydroxybutyrate (Fig. 2D). In fact, serum β-hydroxybutyrate showed a slight increase from 0.69 ± 0.04 mmol/L to 0.90 ± 0.26 mmol/L at 4 h of fasting, reached a maximum of approximately 4 times greater than the baseline level at 72 h of fasting, and then began to decline by 37.5% at 96 h of fasting compared with 72 h of fasting. The increases in serum β-hydroxybutyrate induced by fasting were ameliorated within 1 day of refeeding and remained at low values after 3 and 6 days of refeeding.

Figure 2 Blood biochemical parameters changed during 4-day fasting and subsequent refeeding.

Blood glucose concentration (A), triglycerides (B), free fatty acids (C) and serum β-hydroxybutyrate (D) concentration in rats before fasting (F0h) and fasting for 4, 8, 12, 24, 48, 72 or 96 h (Fxh), and after 1-, 3-, or 6-day refeeding (Rxd) following 96 h fasting. Data were presented as means ± SD (n = 6). One-way ANOVA was used with multiple comparisons using post hoc Tukey’s test. F, Fasting; R, Refeeding. Significance was determined as *, #, $, p < 0.05; **, ##, $, p < 0.01, *, VS, respectively.

Characterization of hormonal parameters in response to fasting and refeeding

The effects of different fasting (n = 6 in each group) and refeeding (n = 6 in each group) durations on the levels of the serum hormones insulin, glucagon, leptin, adiponectin, and FGF21 are shown in Fig. 3. Serum insulin decreased from 5.22 ± 0.86 mU/L to 4.63 ± 1.14 mU/L after 4 h of fasting (Fig. 3A). It continued to decline in the subsequent fasting period, reached its lowest but detectable levels at 72 h (2.72 ± 0.78 mU/L) and even remained at a low level until 96 h of fasting. On the contrary, serum glucagon increased from 30.37 ± 4.08 mU/L to 33.11 ± 3.53 mU/L after 4 h of fasting, and reaching the highest level at 24 h (41.55 ± 2.98 mU/L). It remained at a high level until 48 h of fasting, then declined at 48 h and 96 h (Fig. 3B). The glucagon/insulin molar ratio increased after 4 h of fasting and reached to the highest level at 48 h of fasting, then it remained at high level until 96 h of fasting (Fig. 3C). Serum leptin also notably decreased within 4 h of fasting, reaching the lowest level at 12 h (0.38 ± 0.02 ng/mL), then slightly elevated at 24 h and 48 h (0.59 ± 0.08 ng/mL and 0.61 ± 0.07 ng/mL, respectively), and remained at 0.59 ± 0.03 ng/mL after 96 h of fasting (Fig. 3D). However, serum adiponectin showed no major changes during fasting (Fig. 3E). Refeeding quickly recovered the serum hormone levels. Serum insulin recovered from 2.78 ± 0.49 mU/L to 3.79 ± 0.63 mU/L after 1 day of refeeding and continued to increase at 3 days and 6 days of refeeding (Fig. 3A). Serum glucagon still remained at a high level after 1 day of refeeding, and restored to its baseline level after 6 days of refeeding (Fig. 3B). The glucagon/insulin molar ratio returned to the base line level after 3 days of refeeding and remained at base line level after 6 days of refeeding (Fig. 3C). Similar changes were also found for serum leptin, but the response was different for serum adiponectin, which was significantly decreased from 3.7 ± 0.28 µg/mL to 2.96 ± 0.25 µg/mL after 1 day of refeeding and remained at low levels after 6 days of refeeding when compared with the baseline value. Fasting increased serum FGF21 beginning at 12 h (31.67 ± 2.67 pg/mL), reached its highest level at 72 h (40.94 ± 4.2 pg/mL), and then markedly decreased to 21.78 ± 5.49 pg/mL at 96 h of fasting (Fig. 3F). There were no significant changes with the different refeeding durations.

Figure 3 Hormonal parameters of rats under 4-day fasting and subsequent refeeding.

Serum insulin (A), glucagon (B), glucagon/insulin (C), leptin (D), adiponectin (E), and FGF21 (F) concentration in rats before fasting (F0h) and fasting for 4, 8, 12, 24, 48, 72 or 96 h (Fxh), and after 1-, 3-, or 6-day refeeding (Rxd) following 96 h fasting. Data were presented as means ± SD (n = 6). One-way ANOVA was used with multiple comparisons using post hoc Tukey’s test. F, Fasting; R, Refeeding. Significance was determined as *, #, $, p < 0.05; **, ##, $, p < 0.01, *, VS, respectively.

The content of leptin and adiponectin in visceral and subcutaneous also were assayed. Fasting quickly decreased the contents of leptin in visceral. It decreased from 9.69 ± 1.09 ng/g to 7.66 ± 0.27 ng/g after 4 h of fasting (Fig. S2C), and remained at a low level during the following fasting duration. It reached to the lowest level (5.45 ± 0.65 ng/g) at 72 h of fasting, and slightly raised at 96 h of fasting. One day of refeeding could not restore the leptin levels to the base line, and it recovered from 7.40 ± 1.15 ng/g to 9.28 ± 0.27 ng/g after 3 day of refeeding and continued to increase at 6 days of refeeding. The content of leptin in subcutaneous had no change till 8 h of fasting, and began to decline from 12 h of fasting and reached the lowest level after 72 h of fasting (Fig. S2D). It returned to the base line level after 1 day of refeeding and remained at base line level after 6 days of refeeding. No significant changes were observed of adiponectin in visceral during the fasting and refeeding period (Fig. S2E). Unlike the adiponectin in visceral, adiponectin increased following the fasting duration in subcutaneous (Fig. S2F). It increased from 12 h of fasting and reaching to the highest level at 72 h of fasting (27.66 ± 0.55 µg/g). Refeeding quickly restored the content of adiponectin to the base line level.

Hepatic glucoregulatory, lipid storage, and metabolic responses to fasting and refeeding

Hepatic glucose production is crucial for glucose homeostasis, which mainly relies on either glycogen breakdown or gluconeogenesis from glycerol, amino acids, or tricarboxylic acid (TCA) cycle intermediates during fasting or food deprivation (Geisler et al., 2016). The effects of different fasting (n = 6 in each group) and refeeding (n = 6 in each group) durations on hepatic glucoregulatory, lipid storage, and metabolic parameters are shown in Fig. 4. The hepatic glycogen content presented a significant decrease in the fasting rats and showed a more than 19% decline within 4 h of fasting, reaching its minimum (34.18 ± 2.41 ng/g) at 24 h (Fig. 4A). Hepatic glycogen markedly increased with 1 day of refeeding but was lower than the baseline level. Hepatic glycogen further increased and reached baseline levels after three days of refeeding.

Figure 4 Hepatic glucoregulatory, lipid storage and metabolites responses to fasting and refeeding.

Hepatic glycogen (A), PEPCK activity (B), Triglyceride (C), and Free fatty acid (D) in rats before fasting (F0h) and fasting for 4, 8, 12, 24, 48, 72 or 96 h (Fxh), and after 1-, 3-, or 6-day refeeding (Rxd) following 96 h fasting. Data were presented as means ± SD (n = 6). One-way ANOVA was used with multiple comparisons using post hoc Tukey’s test. F, Fasting; R, Refeeding. Significance was determined as *, #, $, p < 0.05; **, ##, $, p < 0.01, *, VS, respectively.

When hepatic glycogen is mostly consumed, gluconeogenesis plays a pivotal role in maintaining glucose homeostasis. PEPCK, an early rate-limiting enzyme in gluconeogenesis, influenced the hepatic gluconeogenesis rate. Hepatic PEPCK activity increased with the duration of fasting and reached its highest level (0.09 ± 0.01 IU/g) at 24 h of fasting remained at a high level from 24 h to 48 h of fasting and then decreased at 72 h and 96 h of fasting (Fig. 4B). Refeeding did not increase the hepatic PEPCK activity, which decreased after 1 day of refeeding and remained at a similar level after 3 days or 6 days of refeeding compared to baseline (0.06 ± 0.01 IU/g).

However, as glycogen was largely consumed, ketone bodies became the primary energy source. To elaborate on the induction of ketogenesis, we first measured the hepatic accumulation of hepatic TG and FFAs, the primary substrates for ketone synthesis. Hepatic TG markedly decreased from 9.88 ± 1.05 mg/g at baseline to 3.25 ± 0.49 mg/g at 96 h of fasting (Fig. 4C). Refeeding quickly reversed this trend and returned the level to the baseline level within 3 days of refeeding. However, hepatic FFAs significantly increased with the extension of the fasting duration, reached the highest level (2.5 ± 0.2 µmol/g) at 24 h of fasting, and then slowly decreased to the baseline level (1.68 ± 0.18 µmol/g) after 96 h of fasting (Fig. 4D). Hepatic FFAs remained at the baseline level at each refeeding time point.

Expression of genes and proteins related to gluconeogenesis in the liver

During prolonged fasting, gluconeogenesis is the pivotal way to maintain glucose homeostasis. To fully detect the different changes in gluconeogenesis-related genes and proteins in the liver, qRT-PCR and western blotting were used to measure the mRNA and protein expression of two key gluconeogenesis genes (PEPCK and G6Pase) at each fasting (n = 6 in each group) and refeeding (n = 6 in each group) time point. The mRNA level of PEPCK was increased by 28.5% at 12 h of fasting (Fig. 5A), reached the highest level at 24 h of fasting, and then decreased to baseline at 96 h of fasting. In the refeeding period, the mRNA expression of PEPCK continued to decline and remained at a low level. The PEPCK protein (Fig. 5B) significantly increased by 1.75 times at 12 h of fasting compared to the baseline level and reached the highest level (2.23 times) at 24 h. There was a slight decrease from 48 h of fasting until the end of the fasting experiment, but it was also higher than the baseline level. In the following refeeding period, the protein expression of PEPCK decreased by 56.4% (p < 0.01, Fig. 5C) after one day of refeeding compared with that at 96 h of fasting. Hepatic glucose production from gluconeogenesis is also dependent on G6Pase. The mRNA expression of G6Pase increased from 8 h of fasting (10.5%; Fig. 5D) and reached the highest value after 24 h of fasting (71.6%; p < 0.01); then, it was downregulated after 72 h of fasting and remained at a low level until the end of the experiment. Its value remained lower than the baseline level after the subsequent refeeding period.

Figure 5 mRNA and protein expression in hepatic related with gluconeogenesis responses to fasting and refeeding.

Hepatic PEPCK (A) and G6Pase (D) mRNA expression before fasting (F0h) and fasting for 4, 8, 12, 24, 48, 72 or 96 h (Fxh), and after 1-, 3-, or 6-day refeeding (Rxd) following 96 h fasting; PEPCK protein (B) expression before fasting (F0h) and fasting for 4, 8, 12, 24, 48, 72 or 96 h (Fxh); PEPCK protein (C) expression before fasting (F0h) and after 1-, 3-, or 6-day refeeding (Rxd) following 96 h fasting. Data were presented as means ± SD (n = 6) One-way ANOVA was used with multiple comparisons using post hoc Tukey’s test. F: Fasting; R: Refeeding. Significance was determined as *, #, $, p < 0.05; **, ##, $, p < 0.01, *, VS, respectively.

Expression of genes and proteins related to ketogenesis in the liver during fasting and refeeding

In addition to glucose metabolism, the liver has a critical role in lipid homeostasis. When glucose reached its minimum, ketone bodies serve as the main energy sources for tissues in the mid-phase of prolonged fasting. The lipolytic and ketogenic responses partly depend on the expression of PPARα, which promotes the expression of genes essential to enhancing ketogenesis (CPT-1a and HMGCS2). The expression of PPARα mRNA (n = 6 in each group) increased by 53.7% after 8 h of fasting and continued to increase until 48 h (Fig. 6A); then, it was slightly decreased but was still higher than the baseline value. After 96 h of fasting, PPARα expression significantly decreased by 38% after 1 day of refeeding. This trend continued with 45.2% and 58.7% decreases at 3 days and 6 days of refeeding, respectively. CPT-1a mRNA was also significantly elevated at 12 h of fasting and reached its highest value at 48 h of fasting, followed by a slow decrease but remaining higher than that of prefasting (n = 6 in each group; Fig. 6B). Refeeding suppressed CPT-1a mRNA expression. CPT-1a encourages the flux of fatty acids through β-oxidation, which results in the production of acetyl-CoA in the fasted liver. HMGCS2 is then required for the flux of acetyl-CoA into ketogenesis. HMGCS2 catalyzes the first step of ketogenesis and plays a pivotal role in regulating the serum ketone body level. The mRNA expression of HMGCS2 (n = 6 in each group; Fig. 6C) in the liver was highly elevated by 87% after 8 h of fasting and remained at a high level at the subsequent time points (74.5% at 12 h, 133% at 24 h, 137% at 48 h, and 92.5% at 72 h; p < 0.01), while subsequently recovering to the baseline value at 96 h of fasting. HMGCS2 mRNA expression continued to decrease after 1 day of refeeding and remained lower after 3 and 6 days of fasting. The HMGCS2 protein level (n = 4 in each group; Fig. 6D) in the liver markedly increased by 1.68 times from 24 h of fasting and reached the highest value at 96 h of fasting. Refeeding did not significantly decrease HMGCS2 protein levels; HMGCS2 protein levels were slightly decreased but still at a high level (n = 4 in each group; Fig. 6E).

Figure 6 mRNA and protein expression in liver related with ketogenesis responses to fasting and refeeding.

Liver PPAR α (A), CPT-1a (B), HMGCS2 (C) mRNA expression before fasting (F0h) and fasting for 4, 8, 12, 24, 48, 72 or 96 h (Fxh), and after 1-, 3-, or 6-day refeeding (Rxd) following 96 h fasting. HMGCS2 protein expression before fasting (F0h) and fasting for 4, 8, 12, 24, 48, 72 or 96 h (Fxh) (D); and HMGCS2 protein expression before fasting (F0h) and after 1-, 3-, or 6-day refeeding (Rxd) following 96 h fasting (E). Data were presented as means ± SD (n = 6). One-way ANOVA was used with multiple comparisons using post hoc Tukey’s test. F, Fasting; R, Refeeding. Significance was determined as *, #, $, p < 0.05; **, ##, $, p < 0.01, *, VS, respectively.

Hepatic nutrient-sensing signaling molecule responses to fasting and refeeding

The nutrient-sensing signaling molecules AMPK, SIRT1, and mTOR are considered essential energy status sensors and modulators of hepatic metabolism in mammals (Cetrullo et al., 2015; Giovannini & Bianchi, 2017). Hepatic nutrient-sensing signaling molecule responses to fasting (n = 4 in each group) and refeeding (n = 4 in each group) are shown in Fig. 7. For this experiment, we chose 4 groups of data for further analysis. The expression of total AMPK protein was slightly increased after 4 h of fasting and noticeably decreased at 24 h of fasting (Fig. 7B). The proportion of p-AMPK (Thr172) to total AMPK was upregulated during 4 h of fasting, reached a maximum after 24 h, and was slightly decreased at 96 h of fasting (Fig. 7C). In contrast, the protein expression of mTOR was not affected until 72 h of fasting (Fig. 7D). The ratio of mTOR p-Ser2448 to total mTOR showed an increasing trend from 4 h of fasting, but there was no significant difference (Fig. 7E). Similar to AMPK, the expression of SIRT1 was upregulated from 4 h of fasting and remained at a high level until the end of the 4-day fasting period (Fig. 7F). The expression of AMPK protein significantly decreased after 1 day of refeeding but was slightly increased after 3 days of refeeding and remained at a high level after 6 days of refeeding (Fig. 7H). Refeeding did not influence the protein expression of mTOR or the ratio of mTOR p-Ser2448 to total mTOR (Figs. 7J and 7K). During the refeeding period, the expression of SIRT1 was also higher than the baseline level (Fig. 7L).

Figure 7 Nutrient-sensing signal molecules responses to fasting and refeeding.

A is the collection of p-AMPK, AMPK, p-mTOR, mTOR and SIRT1 protein bands before fasting (F0h) and fasting for 4, 8, 12, 24, 48, 72 or 96 h (Fxh). The expression ratio of AMPK (B), mTOR (D), SIRT1(F) to β-actin before fasting (F0h) and fasting for 4, 8, 12, 24, 48, 72 or 96 h (Fxh); The ratio of p-AMPK thr172 to total AMPK protein (C), p-mTOR ser2448 to total mTOR protein (E) were analyzed before fasting (F0h) and fasting for 4, 8, 12, 24, 48, 72 or 96 h (Fxh). G is the collection of p-AMPK, AMPK, p-mTOR, mTOR and SIRT1 protein bands before fasting (F0h) and after 1-, 3-, or 6-day refeeding (Rxd) following 96 h fasting. The expression ratio of AMPK (H), mTOR (J), SIRT1 (L) to β-actin before fasting (F0h) and after 1-, 3-, or 6-day refeeding (Rxd) following 96 h fasting. The ratio of p-AMPK thr172 to total AMPK protein (I), p-mTOR ser2448 to total mTOR protein (K) were analyzed before fasting (F0h) and after 1-, 3-, or 6-day refeeding (Rxd) following 96 h fasting. Data were presented as means ± SD (n = 6). One-way ANOVA was used with multiple comparisons using post hoc Tukey’s test. F: Fasting; R: Refeeding. Significance was determined as *, #, $, p < 0.05; **, ##, $, p < 0.01, *, VS, respectively.

Discussion

Mammals have developed multiple mechanisms to accommodate environmental changes, such as food availability (Newman & Verdin, 2017). As the central organ in the regulation of metabolism, the liver plays an irreplaceable role in maintaining energy and nutrient stability during starvation and refeeding. Moreover, the liver is subjected to various nutrient stresses (Trefts, Gannon & Wasserman, 2017). Here, we showed the detailed timing of the metabolic response to 4-day complete prolonged fasting and refeeding. Metabolism-related indicators and pathways were evaluated, including glycogenolysis, gluconeogenesis, β-oxidation, ketogenesis, hormonal parameters, and nutrient signaling molecules. Fasting quickly consumed glycogen, triggered the energy substrates to shift from glucose to ketones, and induced a series of serum hormone level changes and molecular expression changes in the liver. Furthermore, short-term refeeding reversed most of the existing energy switches.

Usually, glycogen is the main source of serum glucose during the initial stage of food deprivation. Hepatic glycogen significantly decreased within 4 h of fasting and was almost fully exhausted by 24 h, corresponding to a maximal serum glucose drop at 24 h of fasting (Fig. 2). Thereafter, gluconeogenesis was the primary source of endogenous glucose production after hepatic glycogen was depleted. When the fasting duration exceeded 12 h, the mRNA and protein expression levels of the gluconeogenesis rate-limiting enzymes PEPCK and G6Pase increased and reached their highest levels at 24 h of fasting and remained at a high level from 24 h to 48 h of fasting (Fig. 5). Similarly, the same expression pattern was also observed for PEPCK activity (Fig. 4B). These results indicated that gluconeogenesis became the leading source for glucose homeostasis. Our results agreed with Goldstein & Curnow’s (1978) report, which depicted that fasting induced parallel changes in plasma glucose and hepatic glycogen concentrations with decreases by 24 h. Hepatic glycogen was almost exhausted from 72 h to 96 h of fasting; blood glucose level was little elevated than that in 24 h of fasting. At the same time, the mRNA and protein expression levels of liver gluconeogenesis reduced. But the mRNA expression levels of PEPCK and G6Pase in the kidney were significantly increased (data not shown), so the raised blood glucose might be contributed by the kidney gluconeogenesis. Hepatic glycogen significantly increased after 1 day of refeeding and was restored to the baseline level by 3 days of refeeding. At the same time, a decrease in G6Pase and PEPCK mRNA and protein was observed, which indicated that the switch from gluconeogenesis to glycolysis may completely occur within 1 day following refeeding. In contrast, some other research has observed different results in mice with short-term fasting and refeeding. In that study, hepatic glycolysis remained low during the initial refeeding phase, while gluconeogenesis remained active until hepatic glycogen levels were restored (Bulik, Holzhütter & Berndt, 2016). The difference in the results between these two studies was mainly caused by different refeeding periods. This finding also indicated that hepatic glycogen played a central role in maintaining short-term glucose homeostasis during transitions between the fed and fasted state (Youn et al., 2021).

When the liver glycogen was exhausted, lipolysis served as the primary energy source for other tissues, corresponding with the decrease in insulin during prolonged fasting (Lessan & Ali, 2019). Lipolysis was the process by which TGs were hydrolyzed to FFA and glycerol. A transient increase in blood TG occurred at 4 h of fasting, followed by a cliff-like decline (Fig. 2B). However, a marked reduction in hepatic TG was found after 4 h of fasting that reached a minimal value at 24 h. These results were in agreement with the findings of others (Groener & Van Golde, 1977; Herrera & Freinkel, 1968). The blood TG level remained at a lower level after 6 days of refeeding and was slightly higher than that after 96 h of fasting. Nevertheless, hepatic TG rapidly increased after refeeding and was restored to the prefasting level after 3 days of refeeding. These results implied that the hydrolysis of TG was triggered in the initial stage of fasting, and the constant consumption led to a continuous decline in serum TG levels. Stimulation of lipolysis resulted in a profound increase in FFA release from adipose tissue. In line with the enhanced lipolysis, circulating and hepatic FFA flux and serum β-hydroxybutyrate increased to the highest levels from 24 h to 72 h of fasting. However, lipolysis did not decrease the FFA flux in visceral and subcutaneous adipose tissue, and it also increased with the fasting duration, and declined at 48 h and 96 h. These results might cause by FFA cycling, which could reesterificate most FFA back into adipose triglycerides before FFA release into plasma in response to metabolic demands (Kalderon et al., 2000). Previous studies have proven that overexpression of CPT-1a or HMGCS2 increased hepatic β-oxidation and ketone synthesis (Geisler et al., 2019; Gwon et al., 2020), while HMGCS2 knockdown eliminated the fasting-induced rise in serum β-hydroxybutyrate (Hepler et al., 2016). In the present study, the mRNA and protein expression of CPT-1a increased by 1.5 times after 12 h of fasting, while that of HMGCS2 rose from 8 h of fasting until 72 h and then slightly decreased after 96 h of fasting. Thus, maximal changes in mRNA expression induced by fasting might be achieved within 24 h in the rat. According to the hepatic glycogen concentration, the robust elevation of metabolites and the expression of related genes represented a complete induction of the fasting response within 24 h in the rat. FFA flux and triglycerides were significantly decreased within 72 h of fasting and most of metabolism indicators declined to a low level. These subsequent alterations revealed the fuel substrate shift process and indicated that three days of fasting was essential for the metabolic pattern shift induced by fasting.

A previous study proved that several transcription factors control the expression of gluconeogenic, oxidative, and ketogenic enzymes. PPARα plays an essential role in the metabolic adaptation of the liver to fasting situations (Preidis, Kim & Moore, 2017). In this study, the mRNA expression of PPARα in the liver was markedly increased after 8 h of fasting. This change occurred after serum FFA levels increased and circulating insulin levels decreased in response to food deprivation. Furthermore, there was a highly positive correlation between the mRNA expression of PPARα and FFA levels (r = 0.758, p < 0.001; Fig. S1A) but a negative correlation with insulin (r = 0.538, p < 0.001; Fig. S1B). In fact, insulin has been demonstrated to have an inhibitory effect on PPARα mRNA expression in the liver (Guo et al., 2016). Thus, the decrease in insulin could be one of the primary events in the activation of fatty acid oxidation under food deprivation through changes in critical transcription factors in different tissues (Jørgensen et al., 2021). PPARα mRNA expression continued to increase out to 48 h ( p < 0.05, Fig. 6A) and then slightly decreased but remained higher than the baseline level. Highly expressed PPARα signaling limited the potential hepatotoxic effects of lipid accumulation by enhancing lipid oxidation and ketone body synthesis. In our study, there was a highly positive correlation between the mRNA expression of PPARα and CPT-1a and HMGCS2 (r = 0.737 and r = 0.791, p < 0.001, respectively; Figs. S1C and S1D). Thus, PPARα appears to be integral for metabolic adaptations with increased ketogenesis in response to fasting.

Metabolism is controlled by the interaction of many hormones under different nutritional conditions, which maintains the energy reserves necessary to maintain a healthy organism. During periods of fasting, insulin, glucagon, leptin, FGF21, and adiponectin play important metabolic roles. The pancreatic hormones, insulin and glucagon, are critical mediators in coordinating the systemic response to changes in nutritional status, with the liver being a primary site of action. Insulin presented a time-dependent decrease in the rat during the fasting period. It reached the lowest level at 72 h of fasting in the present research, which was consistent with the upregulated gluconeogenesis and glycogenolysis due to its suppressive action (Hatting et al., 2018). Serum glucagon exhibited an opposite trend. It was increased with fasting initiation, and reached to the highest level nearly 48 h of fasting, then it remained at high level till the end of fasting. Previous studies have indicated that glucagon played a critical role in the regulation of hepatic glycogenolysis, gluconeogenesis and fatty acid oxidation during fasting (Stern et al., 2019). The increased serum glucagon in our research proved that the leading to a predominant glucagon signal, which increases hepatic intracellular concentrations of the second messenger cyclic adenosine monophosphate (cAMP) and downstream PKA and CREB signaling pathways, increasing hepatic glucose output (Mutel et al., 2011; Seitz et al., 1977). The serum glucagon/ insulin ratio increased with fasting and decreased upon refeeding were consistent with other research (Mutel et al., 2011; Seitz et al., 1977), and further indicated the reciprocal regulation of glucagon and insulin signal to adapt the fasting and refeeding condition. Most of the fasting-induced changes in metabolite and serum hormone levels were reciprocally altered during refeeding. Indeed, we and other authors have observed increases in insulin and decreases in FFA concentrations in refed rats (Kochan, Karbowska & Swierczyński, 1997). Similar to serum insulin, leptin, which regulates energy homeostasis and modulates satiety by suppressing food intake and stimulating energy expenditure (Zieba, Biernat & Barć, 2020), showed a consistent decreasing trend during fasting and refeeding. Moreover, the content of leptin quickly decreased at the beginning of fasting in visceral but not in subcutaneous. These data supported the view that leptin synthesis responded acutely to changes in nutritional status in a site-specific manner. The earlier and more pronounced decrease in leptin content in visceral compared with subcutaneous would be in line with the knowledge that the mobilization of intra-abdominal fat pads is greater than in subcutaneous during fasting (Shimabukuro et al., 1997; Wang et al., 1998). Adiponectin is an adipokine-specific hormone that is mainly secreted by white adipose tissue and is involved in regulating glucose and lipid metabolism (Nguyen, 2020). Our finding of unaltered serum adiponectin during the fasting and refeeding periods was supported by similar observations in old male rats and mice (Gui, Silha & Murphy, 2004). However, inconsistent of the serum adiponectin in this study, fasting increased the content of adiponectin in subcutaneous, but had no influence on visceral. Similarly, studies in human and rodents had proposed that subcutaneous white adipose tissues were reported to be a major source of adiponectin (Wronska & Kmiec, 2012). Zhang et al. (2002) proved that fasting reduced and refeeding enhanced adiponectin mRNA expression in perirenal adipose tissue of 6-month F344/BN rats, but serum adiponectin was not different from control levels. Moreover, fasting in 8-week-old male SD rats for 1 or 4 days did not change the amounts of adiponectin mRNA in subcutaneous or epididymal adipose tissue (Bertile & Raclot, 2004). Thus, posttranscriptional mechanisms might regulate adiponectin release into the blood and cause different results. FGF21 was described as a key liver-secreted hormonal mediator of the adaptive response to starvation and was attributed to driving ketogenesis (Xiao et al., 2019) and gluconeogenesis (Liang et al., 2014) during starvation. Consistent with others’ research, fasting increased serum FGF21 from 48 h to 72 h but significantly decreased it at 96 h of fasting. This changing trend was similar to that of the mRNA expression of the transcription factor PPARα, which was shown to regulate FGF21 (Inagaki et al., 2007). However, the fasting and refeeding-induced changes in plasma hormone concentrations were also altered when facing different nutrition conditions.

Additionally, several nutrient-sensing signaling molecules (AMPK, SIRT1, and mTOR) are independently or synergistically regulated by multiple pathophysiological processes, such as cell energy metabolism, autophagy, apoptosis, and cell survival in the liver (Giovannini & Bianchi, 2017). As one of the most critical nutrient-regulating signals, mTOR is activated by serine phosphorylation at ser2448 in high-energy states. In this study, the protein expression of mTOR was not affected by the fasting duration. In contrast, the proportion of p-AMPK (Thr172) to total AMPK presented an increasing trend during 4 days of complete fasting. AMPK is the downstream component of a kinase cascade that acts as a sensor of cellular energy status. Once activated, phosphorylated AMPK (Thr172) inhibited not only the expression of the gluconeogenesis-related genes PEPCK and G6Pase but also phosphorylated acetyl-CoA carboxylase (ACC), which led to a switch in the energy-producing pathways that functions as a fuel gauge (Galic et al., 2018). According to the serum glucose and serum FFA levels, the status of AMPK coordinated energy expenditure by controlling gluconeogenesis and fatty acid synthesis during prolonged fasting. The histone deacetylase SIRT1 plays a critical role in protein deacetylation related to cellular metabolism, stress responses, and possibly aging by modulating the activity of transcription factors and cofactors (Wang et al., 2021). SIRT1 functions as a master switch to maintain lipid and glucose homeostasis and energy balance by regulating important metabolic regulators (Estienne et al., 2021). Similar to AMPK, the expression of SIRT1 was upregulated at 4 h of fasting and remained at a high level until the end of the 4-day fasting duration. These results suggested that the liver could quickly (after 4 h of fasting in rats in the present study) respond to the alterations in different nutritional environments by activating a series of metabolic pathway reactions after sensing energy changes. The activated status of p-AMPK reached its highest level at the 24-h fasting time point in this rat fasting experiment, which to some extent implied that the liver could produce and maintain a new homeostasis after 24 h of fasting in rats. Besides the nutrient-sensing signaling molecules, the cytoplasmic and mitochondrial redox state and hepatic level of energy charge (EC) also inflects the energy metabolism response to fasting and refeeding. These parameters, to some extent, influence the changes of nutritent-sensing signaling molecules. Previous studies have proven that phosphorylated AMPK (Thr172) is sensitive to cellular AMP/ATP ratio (Stein et al., 2000). Adenine nucleotides (ATP, ADP, and AMP) usually is used to evaluate liver energy charge (Díaz-Muñoz et al., 2000). Start & Newsholme have reported that rats fasted for 24 h show a decrease in hepatic ATP, total adenine nucleotides, and energy charge levels (Start & Newsholme, 1968). Liver energy metabolism is complex, the present research differs from these observations in several aspects. However, comparison between both studies is difficult due to a difference in experimental design. A more comprehensive design is required to fully understand the changes in liver energy sensing factors during fasting and refeeding.

Actually, except the regulatory process of the body’s own homeostasis during fasting-feeding, circadian rhythms also play an important role to allow animals to coordinate behavioral and physiological processes with respect to one another and to synchronize these processes (Dreyer et al., 2019). Previous studies have proved that the blood glucose level, the glycogen concentration, in fed and starved rats displayed rhythmic oscillations with a 24- or 12-hour period in the course of the day, but one day’s fasting or prolonged fasting do not affect the character of circadian rhythm (Ahlersová et al., 1980; Marliss et al., 1970). What’s more, the presence of a food-entrainable oscillator (FEO), which is independent from the SCN, is relate to metabolic rhythms (Escobar et al., 1998). In this research, we concern a more detail fasting duration from 4 h to 96 h, and almost each sacrifice time is on the light time, therefore less affected by circadian rhythms. Despite this, rhythm factor is an important influencing factor of intermittent fasting, so it necessary to be further verified in-depth research.

Given that many of the metabolic pathways are active and serve metabolic roles during prolonged fasting, careful consideration must be applied to study the design of prolonged intermittent fasting regimens.

Conclusions

Taken together, these results show that there were changes in the metabolism of glucose and lipids based on the timing of fasting and refeeding. The results showed that glycogen was an essential source of glucose during 24 h of fasting, while lipids were mobilized during prolonged fasting. A significant energy metabolic substrate switch from glucose to ketone bodies occurred between 24 h and 72 h of fasting. One day of refeeding quickly recovered this metabolic switch after 4 days of fasting. The metabolic activity of the liver is involved in satisfying the energy needs during periods of restricted fasting and feeding. This research highlights the timing of the metabolic adaptation changes in response to prolonged fasting and refeeding, which may provide a reference for designing a reasonable intermittent fasting interval to prevent adverse effects. However, there are limitations to this study. The detailed series of changes in the different hormones, energy-sensing factors, and blood parameters in response to fasting and refeeding are not fully understood. The relation between circadian systems and fasting and refeeding is lacking and further research with larger samples is necessary to validate the different sequences of changes and circadian fluctuations during prolonged fasting and refeeding.

Supplemental Information

Supplemental Information 1 Raw data

Click here for additional data file.

Supplemental Information 2 Supplemental figures and data

Click here for additional data file.

Figure S1 The correlations between the expression of PPAR α and other factors

(A) FFA (µmmol/L); (B) insulin (mU/L); (C) the expression of HMGCS2 and (D) the expression of CPT-1a.

Click here for additional data file.

Figure S2 The content of FFA, Leptin, and Adiponectin in visceral adipose tissue and subcutaneous adipose tissue

VAT: visceral adipose tissue. SAT: subcutaneous adipose tissue. (A) FFA in VAT; (B) FFA in SATs; (C) Leptin in VAT; (D) Leptin in SAT; (E) Adiponectin in VAT; (F) Adiponectin in SAT.

Click here for additional data file.

Supplemental Information 5 Author Checklist

Click here for additional data file.

Additional Information and Declarations

Competing Interests

Author Contributions

Animal Ethics

Data Availability

The authors declare there are no competing interests.

Xiukun Sui conceived and designed the experiments, performed the experiments, analyzed the data, prepared figures and/or tables, authored or reviewed drafts of the article, and approved the final draft.

Hailong Wang conceived and designed the experiments, performed the experiments, analyzed the data, prepared figures and/or tables, authored or reviewed drafts of the article, and approved the final draft.

Feng Wu performed the experiments, prepared figures and/or tables, and approved the final draft.

Chao Yang analyzed the data, prepared figures and/or tables, and approved the final draft.

Hongyu Zhang performed the experiments, prepared figures and/or tables, and approved the final draft.

Zihan Xu analyzed the data, prepared figures and/or tables, and approved the final draft.

Yaxiu Guo performed the experiments, analyzed the data, prepared figures and/or tables, and approved the final draft.

ZhiFeng Guo performed the experiments, prepared figures and/or tables, authored or reviewed drafts of the article, and approved the final draft.

Bingmu Xin performed the experiments, authored or reviewed drafts of the article, and approved the final draft.

Ting Ma conceived and designed the experiments, authored or reviewed drafts of the article, and approved the final draft.

Yinghui Li conceived and designed the experiments, authored or reviewed drafts of the article, and approved the final draft.

Zhongquan Dai conceived and designed the experiments, analyzed the data, prepared figures and/or tables, authored or reviewed drafts of the article, and approved the final draft.

The following information was supplied relating to ethical approvals (i.e., approving body and any reference numbers):

The Committees of Animal Ethics and Experimental Safety of the China Astronaut Research and Training Center approved the study (ACC-IACUC-2021-030).

The following information was supplied regarding data availability:

The raw data is available in the Supplemental File.

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
