# Peer review of "Hepatic metabolite responses to 4-day complete fasting and subsequent refeeding in rats"

_PeerJ, doi:10.7717/peerj.14009_

## Round 0.1 · original submission · Major Revisions

Having read the manuscript, I believe that the English language needs to be improved before it can be sent out for peer review. There is an unacceptably high number of errors, particularly grammatical errors.

I strongly recommend that a professional English language editing service is used to bring the manuscript up to the required standard.

Once that has been done, I will be happy to send the article out for formal peer review.

---

## Round 0.2 · Major Revisions

Both reviewers has noted some major issues that need to be addressed in a revised version of the manuscript.

Reviewer 1 notes that there is a lack of novelty in the results. While the lack of novelty does not mean that the work shouldn't be publishable, it would help if the authors can clearly articulate how their research adds to the field, or clearly acknowledge if it is confirming other data.

The English language is much improved compared with the previous version, however, there are still some errors that should be addressed.

Reviewer 1 ·

Basic reporting

Sui et al, reported a set of experiments exploring the liver response to protocols of prolonged fasting and refeeding. Their data involved determinations of metabolites, presence of enzymes and hormones as well as some nutrient sensors.

My main criticism is that the manuscript contributes very little with original or innovative results. Indeed, metabolic, molecular and endocrine responses to fasting and refeeding are extensively reported (please checked numerous review publications in PubMed). To my point of view, Sui et al., did not find any result that could provide a significant advance to the field of nutritional physiology.

Major issues

1. The approach of the authors is to relate their fasting / refeeding protocol with nutritional situations such as intermittent fasting. However, it is necessary to consider that the authors` study is an acute approach. Indeed, with several times of fasting and several times of refeeding, but at the end, it is only one episode of food removing and food reintroduction. In contrast, intermittent fasting protocols involve a semichronic or chronic condition. Physiological adaptations in these 2 conditions are very different (Pérez-Mendoza et. al, 2014, Chronobiol Int.31, 815).

2. One conceptual weakness of this study is not to consider that fasting / refeeding protocols act as circadian synchronizer. As a matter of fact, intermittent fasting protocols in experimental systems are used to characterize the expression of the food entrained oscillator (FEO) (Escobar et al., 1998, Am J Physiol, 274, R1309). The manuscript of Sui et al, would benefit if they incorporate in the Introduction and mainly in the Discussion some concepts of circadian physiology.

3. The authors found in several of the fasting parameters studied, that the more clear changes were obtained the first 24/48 h. Next times, 72 and 96 h the changes became more discreet. The authors failed to discuss the implications of this consistent pattern.

Experimental design

No comments

Validity of the findings

Some considerations to strengthen the Sui et al, report:
1. Even though the study is focused in the liver, several parameters are mainly related to the adipose tissue (FFA, leptin and adiponectin). It would be very useful to measure the presence and variations of adipose tissue (visceral, subcutaneous, gonadal).

2. Because the rhythmic implications of this study, it is convenient to say the time of the day that the food is removed and reintroduced, as well as the time of the animals´ sacrifice.

3. It is necessary to mention the low levels of glucose and high levels of beta-OHBut that were considered as dangerous for the experimental animals.

4. In the Introduction section some concepts regarding FGF21 are absent.

5. It is very informative to measure both pancreatic hormones, insulin and glucagon, since these hormones act as with an opposite nutritional role (Martiñón-Gutiérrez et al., 2021, Sci Rep, 11, 11666.

6. Extra caution should be taken when liver is removed for molecular experiments when blood has not been completely washed.

7. Even though the presence of AMPK, mTOR and SIRT1 are informative, a more complete consideration of their metabolic activity is obtained when parameters such as energy charge and redox state are determined or at least discussed (Díaz-Muñoz et al., 2000, Am J Physiol, 279, R2048).

Additional comments

No comments

·

Basic reporting

In this study, Sui and colleagues detail the incremental metabolic response and hepatic adaptation in response to a 4-day fast and refeeding in rats. The authors conclude that this study provides insight and information for the design of prolonged and intermittent fasting regimens.
This reviewer found the study to be a comprehensive and an informative description of the metabolic response to incremental fasting duration with subsequent re-feeding, focusing on the hepatic response. I have some concerns that need to be addressed:
Data analyses and presentation:
1. Please provide individual data points within bar graphs.
2. Statistical analyses for all longitudinal data should be performed using repeated measures ANOVA, not t-tests.
3. Statistical analyses for cross-sectional analyses should be performed using ANOVAs with post hoc analyses to correct for multiple comparisons and determine where differences between groups lie. Please change accordingly and use superscripts to indicate statistical significance.
4. On that note, as the data is currently plotted in figures 1-4, it is difficult to ascertain whether data is longitudinal or in different animals. If each time group is a separate group, a line graph should not be used. Please change accordingly to bar graphs where applicable.
5. Please provide raw data/ original blots for western blots
Discussion:
1. The discussion is well written and includes a thorough description of the literature. However, while the authors provide good description of the role of insulin signaling in fasting and re-feeding, they did not discuss the role of the counterregulatory hormone glucagon, which plays a critical role in the metabolic pathways the authors have investigated. This reviewer would appreciate the inclusion of some discussion on the role of hepatic glucagon signaling in glycogenolysis, gluconeogenesis, and the maintenance of glucose homeostasis.
2. There are some minor issues with grammar and tense. The author goes back and forth between past and present tense- please correct for continuity.

Experimental design

No comment

Validity of the findings

1. Please provide individual data points within bar graphs.
2. Statistical analyses for all longitudinal data should be performed using repeated measures ANOVA, not t-tests.
3. Statistical analyses for cross-sectional analyses should be performed using ANOVAs with post hoc analyses to correct for multiple comparisons and determine where differences between groups lie. Please change accordingly and use superscripts to indicate statistical significance.

---

## Round 0.3 · accepted · Accept

The authors have made significant changes to their manuscript, and by doing so I believe they have appropriately addressed the comments raised by both reviewers. The only outstanding issue from my point of view is that there are still some grammatical and typographical errors throughout the manuscript, and I therefore encourage the authors to undertake a final careful check of the manuscript to address these errors.